# The relationship between anemia and sleep disturbances among older Chinese adults: The mediating role of handgrip strength

**Jie Li[1]\*, Zunyi Ma[2], Xiaojiang Zhao[3]\***

1 Sports Department, Shanxi Politics and Law Institute for Administrators, Taiyuan, China, 2 Cultural Tourism College, Sichuan Cultural Industry Vocational College, Chengdu, China, 3 Department of Physical Education and Arts, Bengbu Medical University, Bengbu, China

\* lijie10242024@163.com (JL); zhaoxiaojiang2010@163.com (XZ)

## Abstract

### Background

Sleep plays an important role in human health, and anemia can lead to a continuous deterioration of sleep. However, the association and mechanism between anemia and sleep disturbances remain unclear. This study aimed to examine the relationship between anemia and sleep disturbances among older Chinese adults, in addition to examining the mediating role of handgrip strength in this relationship.

### Methods

This research utilized data from the 2015 China Health and Retirement Longitudinal Study (CHARLS). Following the application of specific inclusion and exclusion criteria, a total of 6,057 Chinese adults aged 60 and above were finally selected as the analysis samples. The dependent variable was sleep disturbances (yes/no), with anemia (present/absent) serving as the main independent variables. Handgrip strength was employed as the mediating factor. Multivariable-adjusted logistic regression analyses were conducted to investigate the relationships among anemia, handgrip strength, and sleep disturbances. Additionally, bootstrap tests were performed to assess the mediating role of handgrip strength in the association between anemia and sleep disturbances.

### Results

In the unadjusted model, a positive association was observed between anemia and sleep disturbances (OR: 1.22, 95% CI: 1.08–1.38), whereas handgrip strength exhibited a negative association with sleep disturbances (OR: 0.95, 95% CI: 0.93–0.97). These associations persisted even after adjusting for covariates in Models 1, 2, and 3. Furthermore, handgrip strength was found to significantly mediate the link between

**Data availability statement:** All relevant data are within the manuscript and its Supporting Information files.

**Funding:** This work was supported by the Philosophy and Social Sciences Foundation of the Anhui Higher Education Institutions of China (2024AH052821) and Philosophy and Social Sciences Foundation of the Anhui Higher Education Institutions of China (2024AH0052823).The funders had no role in study design, data collection and analysis, decision to publish, or preparation of the manuscript.

**Competing interests:** The authors have declared that no competing interests exist.

anemia and sleep disturbances(mediating effect = 5.75x10$^{-3}$), with the mediating effect accounting for 15.67% of the association.

## Conclusions

The study indicate that anemia is positively associated with an increased risk of sleep disturbances among older Chinese individuals, with handgrip strength acting as a significant mediator in this relationship. This study provides valuable references for improving the sleep quality of older adults.

---

## 1. Introduction

Quality sleep plays a key role in ensuring good physical and mental health [1]. With the accelerating pace of modern life, an increasing number of individuals are experiencing varying degrees of sleep disturbances, the prevalence of which rises with age [2]. Sleep disturbances are defined as self-reported poor sleep quality, clinical insomnia, and/or changes or deficiencies in sleep parameters [3].Sleep disturbances have emerged as a global health concern in contemporary society [4]. A meta-analysis examining the general populations of the Netherlands, the United Kingdom, and the United States found that the prevalence of sleep disturbances ranges from 9.6% to 19.4% [5]. Furthermore, another research shows that 47.2% of Chinese individuals over 60 experience sleep disturbances [6,7]. Sleep disturbances can cause cognitive impairment, reduced alertness, mood deterioration, cardiovascular illnesses, and even lead to death [8]. Consequently, investigating the mechanisms underlying sleep disturbances and their treatment is of paramount importance. Anemia is characterized by a red blood cell count or hemoglobin level that is lower than normal in the peripheral blood [9,10]. Which can lead to fatigue and other health issues, which may in turn affect sleep quality and duration. Several studies have shown that anemia significantly affects sleep quality in older adults [11–13]. For instance, a study suggested a complex interplay between sleep, metabolic health, and hemoglobin levels [14]. Another study shows that orexin deficiency may aggravate the symptoms of anemia and thereby may lead to sleep disturbances [15]. Furthermore, the relationship between sleep disturbances and nutritional deficiencies, is explored in a study focusing on excessive daytime sleepiness (EDS) in older adults [16]. Anemia are affected by various factors, including sex, age, and smoking status [17]. Due to the challenges associated with gathering extensive data from a large cohort, the precise correlation between anemia and sleep disturbances within the general population remains unclear. Handgrip strength has been shown to be predictive of major health-related events in older persons [18]. Recent evidence indicates a significant link between handgrip strength and sleep disturbances. Better handgrip strength in older adults is associated with improved sleep quality. For example, a study using data from the United States from 2005 to 2018 indicated that reduced grip strength was associated with an increased probability of experiencing sleep disturbances [19,20]. A separate investigation examined the causal

relationships between sleep characteristics and diminished grip strength utilizing a Mendelian randomization methodology. This research identified a positive association between specific sleep disturbances, including sleep-wake disorders, and reduced grip strength [21]. In contrast, one study found no link between handgrip strength and sleep disturbances [22]. In addition, anemia may further exacerbate decreased grip strength and sleep disturbances by affecting oxygen transport and muscle metabolism. Insufficient oxygen supply caused by anemia may affect the function and recovery ability of muscles, thereby leading to weakened grip strength. At the same time, anemia may also lead to a decline in sleep quality by affecting the oxygen supply to the brain [23]. This highlights the need for further studies. Overall, previous research has identified independent associations between anemia and sleep disturbances, as well as between grip strength and sleep disturbances. However, the potential mediating role of grip strength in the relationship between anemia and sleep disturbances has not yet been investigated, particularly among older Chinese adults. Therefore, this study utilizes data from the China Health and Retirement Longitudinal Study (CHARLS) to investigate the association between anemia, handgrip strength, and sleep disturbances, and to explore the potential mediating role of handgrip strength in this relationship. This conceptual framework illustrates the relationship between anemia (independent variable), handgrip strength (mediating variable), and sleep disturbances (dependent variable), as depicted in S1 Fig.

## 2. Methods

### 2.1 Data source

The data comes from CHARLS. CHARLS is a longitudinal survey that represents the national population of Chinese individuals aged 45 and above living in communities. The survey aims to establish a comprehensive, publicly accessible microdata set that is both reliable and representative. CHARLS employed a multistage, stratified sampling method, utilizing probability proportional to size, to select 450 primary sampling units from 150 counties or districts. Detailed descriptions of the CHARLS survey methodology have been provided in previous studies [24]. The CHARLS study received approval from the Peking University Ethics Committee (IRB00001052–11015), and all participants provided informed written consent. In our study, we relied on data collected in 2015 (Wave three). The initial cohort comprised 21,095 respondents. To ensure analytical rigor, exclusion criteria were applied sequentially: individuals younger than 60 years (n = 11,139), those with unrecorded sleep disturbances data (n = 885), participants lacking anemia information (n = 2,719), individuals with incomplete handgrip strength assessments (n = 165), and those missing covariate information (n = 128). After applying these exclusions, a total of 6,057 participants remained for analysis. Fig 1 depicts the study population selection process.

### 2.2. Assessment of sleep disturbance, anemia, and handgrip strength

This study utilized the CHARLES questionnaire to diagnose sleep disturbances, focusing on sleep-related issues relevant to the research. Sleep quality assessment was based on responses to the following questions: "My sleep was restless". Participants self-reported sleep quality using these options: rarely or none (<1 day), some or a little (1–2 days), occasionally or moderately (3–4 days), and most or all of the time (5–7 days). Previous researchers' achievements were utilized as points of reference, sleep quality was classified as 'satisfactory' if disturbances occurred rarely or not at all (less than 1 day per week), otherwise as 'unsatisfactory' [25–27]. 'unsatisfactory' sleep quality was defined as the presence of a sleep disturbance, 'satisfactory' sleep quality was defined as the Non-sleep disturbance.

The CHARLS project collaborated with the Chinese Center for Disease Control and Prevention (China CDC) to facilitate the collection and processing of blood samples. For each participant, three tubes of blood were obtained. One of these tubes was promptly stored at 4°C and transported to the nearest CDC or health center for a complete blood count analysis, with a median time from collection to analysis of 97 minutes. The remaining two tubes were preserved at −80°C for subsequent bioassay analysis at a nationally certified laboratory at Capital Medical University [28]. We adapted the

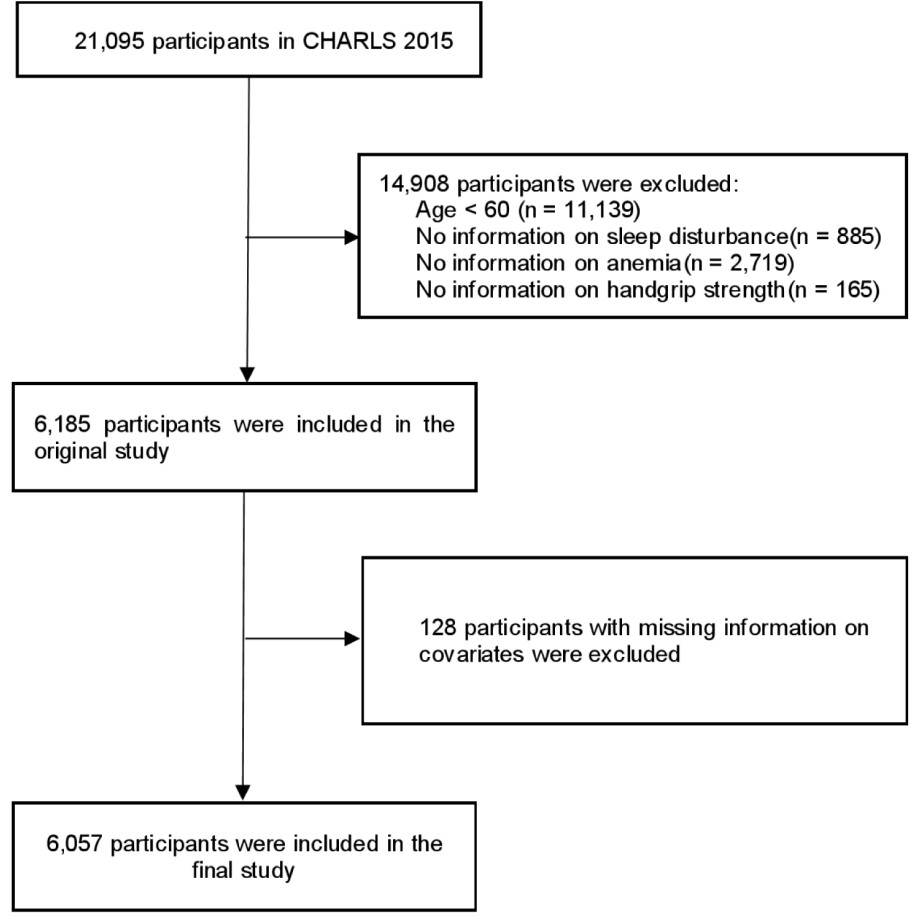

**Fig 1. Flowchart of the participants selection process.** Abbreviations: CHARLS, China health and retirement longitudinal study.

WHO threshold to define anemia as our standard reference, anemia is defined by hemoglobin levels below 12.0 g/dL for women and 13.0 g/dL for men [29]. While WHO cutoffs provide standardized criteria, we acknowledge these do not account for age-related hemoglobin declines.

Handgrip strength was assessed utilizing the Yuejian™ WL-1000 dynamometer (Nantong Yuejian Physical Measurement Instrument Co., Ltd., Nantong, China), with results recorded in kilograms [30]. Participants were instructed by CHARLS testers to stand and hold the dynamometer at a 90° angle, maintaining the position while exerting force on the handle for a duration of 3–5 seconds. Each participant completed two measurements for both the left and right hands, resulting in a total of four measurements. Verbal encouragement was provided to motivate participants to exert maximum effort during the assessments [31]. The highest value obtained from the four measurements, irrespective of whether it was from the left or right hand, was used for analysis.

## 2.3. Covariates

The baseline investigation evaluated a range of demographic and health-related variables, including age, sex (categorized as female or male), geographic residency (categorized as urban or rural), and marital status, categorized as: married and living together, married but living apart, single, divorced, or widowed. The study incorporated additional

covariates such as the count of chronic conditions (categorized as none, one, or two or more), smoking status (smoker or non-smoker), drinking status (classified as non-drinker, occasional drinker, or regular drinker), BMI(underweight, normal weight, overweight, and obese) and educational attainment (grouped as elementary school or below, or middle school or above). The prevalence of chronic diseases was ascertained through self-reported diagnoses of fourteen non-communicable conditions, including hypertension, diabetes, dyslipidemia, chronic pulmonary disease, hepatic disorders, renal disease, cardiovascular events, malignancies, arthritis, asthma, gastrointestinal ailments, cognitive impairment, mental health conditions, and musculoskeletal disorders. Sleep duration were evaluated by inquiring participants about their average nightly sleep duration over the past month.Based on their responses, sleep duration was recorded in hours [32]. Daytime napping was evaluated by asking, "What was your typical post-lunch nap duration in the past month?" Daytime napping was classified into five categories: 0 minutes, less than 30 minutes, 30–59 minutes, 60–89 minutes, and 90 minutes or more [33].

### 2.4. Statistical methods

The study employed R version 4.3.3 for all statistical analyses. Descriptive statistics were utilized to summarize the data. A Shapiro-Wilk statistical test was used to confirm whether or not continuous variables have a normal distribution. Continuous variables with normal distribution were expressed as mean ± SD and contrasted through a one way analysis of variance, and categorical variables by frequencies and percentages. Utilizing the Variance Inflation Factor (VIF) method to assess multicollinearity, a VIF value of five or greater suggests the presence of multicollinearity [34]. According to this criterion, none of the variables exhibit multicollinearity in either sex, as demonstrated in S1 Table. A multivariable logistic regression analysis examined the associations between anemia, handgrip strength, and sleep disturbance, quantifying these associations with odds ratios (OR) and 95% confidence intervals (CI). Four progressively adjusted models were applied: an unadjusted model; Model 1, which was adjusted for age, sex, educational level, and marital status; Model 2, which included the adjustments of Model 1 with additional adjustments for smoking status, drinking status, sleep duration, daytime napping behavior, and body mass index; and Model 3, which was adjusted for all covariates. Subsequently, employing the adjustment variables from Model 3, restricted cubic spline (RCS) fitting was utilized to investigate potential non-linear relationships between handgrip strength and sleep disturbance. Subgroup analysis was then performed. Finally, a mediation effect model was developed to evaluate the mediating role of handgrip strength in the association pathway between anemia and sleep disturbance. The mediating, total, effect's significance was assessed using a bootstrap resampling method with 1,000 iterations [35]. Statistical significance was defined as p-values less than 0.05.

## 3. Results

### 3.1. Characteristics of participants

Table 1 presents the baseline characteristics of the study participants, categorized by sleep disorder status. The study comprised a total of 6,057 participants, of whom 2,959 (48.9%) reported experiencing sleep disorders. The mean age of the participants was 67.7 ± 6.2 years. The average grip strength among the participants was 26.9 ± 9.2 kilograms. The analysis revealed that participants who were female, resided in urban areas, were unmarried, had a lower educational attainment (primary school or below), smoked, consumed alcohol more than once a month, had 14 or more chronic diseases, and were anemic were significantly more likely to experience sleep disturbances (p < 0.001). Additionally, shorter sleep duration, reduced daytime nap time, and lower grip strength were also associated with a heightened risk of sleep disturbances (p < 0.001). Moreover, the analysis of baseline characteristics of included and excluded participants is shown in S2 Table.

**Table 1. General characteristics of study participants according to sleep disturbance(Yes or No).**

| Variables | Total | No | Yes | p value |
|---|---|---|---|---|
| | n = 6057 | n = 3098(51.1) | n = 2959(48.9) | |
| Age, Mean ± SD | 67.7 ± 6.2 | 67.6 ± 6.2 | 67.8 ± 6.3 | 0.445 |
| Sex, n (%) | | | | < 0.001 |
| Female | 3020 (49.9) | 1257 (40.6) | 1763 (59.6) | |
| Male | 3037 (50.1) | 1841 (59.4) | 1196 (40.4) | |
| Residence, n (%) | | | | 0.006 |
| Rural | 3766 (62.2) | 1874 (60.5) | 1892 (63.9) | |
| Urban | 2291 (37.8) | 1224 (39.5) | 1067 (36.1) | |
| Marital status, n (%) | | | | < 0.001 |
| Married and living with a spouse | 4726 (78.0) | 2501 (80.7) | 2225 (75.2) | |
| Married but living without a spouse | 143 (2.4) | 65 (2.1) | 78 (2.6) | |
| Single, divorced, and windowed | 1188 (19.6) | 532 (17.2) | 656 (22.2) | |
| Education Status, n (%) | | | | < 0.001 |
| Elementary school or below | 4830 (79.7) | 2376 (76.7) | 2454 (82.9) | |
| Middle school or above | 1227 (20.3) | 722 (23.3) | 505 (17.1) | |
| Smoking Status, n (%) | | | | < 0.001 |
| Non-smoker | 2874 (47.4) | 1672 (54) | 1202 (40.6) | |
| Smoker | 3183 (52.6) | 1426 (46) | 1757 (59.4) | |
| Drinking Status, n (%) | | | | < 0.001 |
| Drink but less than once a month | 454 (7.5) | 240 (7.7) | 214 (7.2) | |
| Drink more than once a month | 4051 (66.9) | 1956 (63.1) | 2095 (70.8) | |
| Non-drinker | 1552 (25.6) | 902 (29.1) | 650 (22) | |
| BMI group, n (%) | | | | 0.177 |
| Underweight | 447 (7.5) | 214 (7) | 233 (8) | |
| Normal | 3589 (60.1) | 1816 (59.4) | 1773 (60.8) | |
| Overweight | 1637 (27.4) | 869 (28.4) | 768 (26.3) | |
| Obesity | 301 (5.0) | 157 (5.1) | 144 (4.9) | |
| Sleep duration(Hrs), Mean ± SD | 6.3 ± 2.1 | 7.0 ± 1.8 | 5.5 ± 2.1 | < 0.001 |
| Daytime napping duration(Min), Median (IQR) | 30.0 (0.1, 60.0) | 30.0 (0.1, 60.0) | 20.0 (0.1, 60.0) | < 0.001 |
| 14 chronic conditions, n (%) | | | | < 0.001 |
| 0 | 1152 (19.0) | 732 (23.6) | 420 (14.2) | |
| 1 | 1353 (22.3) | 782 (25.2) | 571 (19.3) | |
| ≥2 | 3552 (58.6) | 1584 (51.1) | 1968 (66.5) | |
| Anemia, n (%) | | | | < 0.001 |
| No | 4745 (78.3) | 2479 (80) | 2266 (76.6) | |
| Yes | 1312 (21.7) | 619 (20) | 693 (23.4) | |
| Handgrip strength(kg), Mean ± SD | 26.9 ± 9.2 | 28.5 ± 9.2 | 25.2 ± 8.9 | < 0.001 |

Independent-sample t tests for continuous variables and chi-square tests for categorical variables were used. Abbreviations: BMI, body mass index; Hrs, hours; Min, minutes.

### 3.2. Associations between anemia, handgrip strength, and sleep disturbance

Table 2 presents the multivariable logistic regression analysis results on the link between anemia, handgrip strength, and sleep disturbance. The unadjusted model revealed that older adults with anemia were at a higher risk of sleep disturbances compared to those without anemia (OR 1.22, 95% CI 1.08–1.38). A significant independent negative association

**Table 2. Associations of anemia and handgrip strength with sleep disturbance.**

| Variables | No | Unadjusted | Model 1 | Model 2 | Model 3 |
|---|---|---|---|---|---|
| | | OR(95% CI) | OR (95% CI) | OR(95% CI) | OR (95% CI) |
| Anemia | 6057 | | | | |
| No | 4745 | 1(Ref) | 1(Ref) | 1(Ref) | 1(Ref) |
| Yes | 1312 | 1.22(1.08~0.38) | 1.05(1.01~1.30) | 1.15(1.01~1.32) | 1.17(1.02~1.34) |
| Handgrip strength | 6057 | 0.95(0.96~0.97) | 0.98(0.97~0.99) | 0.98(0.97~0.99) | 0.98(0.97~0.99) |

Model 1: adjusted for age, gender, educational level, marital status, and residence. Model 2: adjusted for model 1 + smoking status, drinking status, sleep duration, daytime napping duration, and BMI. Model 3: adjusted for model 2 + 14 chronic diseases. Abbreviations: OR, odds ratio; 95% CI, 95% confidence interval.

was identified between handgrip strength and the risk of sleep disturbance in the unadjusted model (OR: 0.95, 95% CI: 0.96–0.97), indicating that as handgrip strength increased, the incidence of sleep disturbances decreased(For every additional 1 kg of handgrip strength with sleep disturbances risk is reduced by 5%). Additionally, S3 Table shows that in the unadjusted model, anemia is negatively correlated with handgrip strength (β: −2.95, 95% CI: −2.95(−3.50–-2.40). The association also remained constant when covariates were considered in Models 1, 2, and 3. The RCS of the association between handgrip strength and sleep disturbance is shown in S2 Fig. There was an absence of non-linearity (nonlinearity: p = 0.88).

### 3.3. Subgroup analyses

In the subgroup analysis, after controlling for all covariates, S4 Table demonstrated that the findings were consistent with the primary analysis, suggesting a positive association between anemia and sleep disturbance. However, no significant interaction was observed among the groups(all P for interaction >0.05). Similarly, S5 Table revealed consistency with the main analysis, indicating a negative association between handgrip strength and sleep disturbance. Again, no significant interaction was detected among the groups (all P for interaction >0.05).

### 3.4. The mediating role of handgrip strength in the relationship between anemia and sleep disturbance

The mediation pathway model is presented in Fig 2. After adjusting for control variables, anemia was found to be significantly associated with sleep disturbance ($\beta_0 = 3.67 \times 10^{-2}$, P = 0.016) and handgrip strength ($\beta_1 = -1.36$, P < 0.001). Furthermore, handgrip strength showed a significant association with sleep disturbance ($\beta_2 = -4.23 \times 10^{-3}$, P < 0.001). Bootstrap analysis further indicated that the indirect mediating effect through handgrip strength was $5.75 \times 10^{-3}$ (P < 0.001). The results show that handgrip strength has a potential mediating effect in the relationship between anemia and sleep disturbance, accounting for 15.67% of the total effect variance.

## 4. Discussion

This study examines how handgrip may mediate the relationship between anemia and sleep disturbances in a cohort of older Chinese adults. The study reveals a positive correlation between anemia and sleep disturbance, whereas handgrip strength shows a negative association with sleep disturbance. Consistent with our hypotheses, handgrip strength partially may mediate the relationship between anemia and sleep disturbance.

### 4.1. The correlation between anemia and sleep disturbances

In line with earlier studies [36], the results of this study indicated a positive relationship between anemia and sleep disturbance. A research has shown that sleep disturbances are common among individuals with anemia, particularly in specific

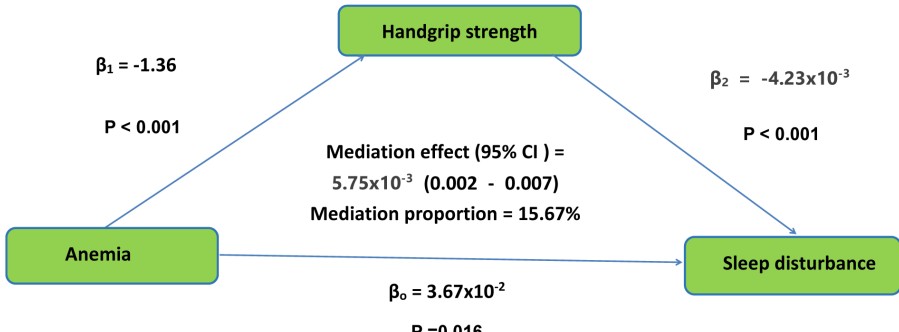

**Fig 2. The conceptual framework of the mediation models.** $\beta_0$ was the total effect of anemia on sleep disturbance; $\beta_1$ represents the effect of anemia on handgrip strength; $\beta_2$ represents the effect of handgrip strength on sleep disturbance. The mediation effect was computed as the product of "$\beta_1$" and "$\beta_2$"($\beta_1 \times \beta_2$), and the mediation proportion was calculated as the ratio of the mediation effect product to total effects [($\beta_1 \times \beta_2$)/$\beta_0$].

populations. This suggests that the presence of anemia may exacerbate sleep-related issues, and addressing sleep-disordered breathing could potentially alleviate some symptoms associated with anemia in these patients [37]. A plausible explanation for the observed association between lower hemoglobin levels and sleep disturbances may involve iron deficiency, although this variable was not assessed in the current study. Notably, body iron status could act as a potential confounding factor in this relationship, rather than a mediating variable, given its established role in both hematological and neurobehavioral processes. The significance of body iron status in emotional behavior and mental health has been documented, as it influences energy metabolism and neurotransmitter homeostasis within the central nervous system [38]. Previous animal studies have demonstrated that iron is crucial for brain myelination, monoaminergic systems, and the homeostasis of glutamate and γ-aminobutyric acid [39]. Consequently, iron deficiency can lead to various cognitive, emotional, and psychological issues due to the disruption of iron-dependent neurochemical pathways. Increased fearfulness has been observed in both animals and humans with inadequate iron status [40]). A recent analysis of a nationwide database from Taiwan indicated a association between iron deficiency and several mental health issues, including anxiety, depression, and sleep disturbances. Furthermore, the study found that iron supplementation therapy was associated with a reduced risk of these psychiatric conditions [41].

## 4.2. The correlation between handgrip strength and sleep disturbances

Handgrip strength is widely recognized as a simple and affordable tool for risk assessment [42]. Prior research examining the relationship between handgrip strength and sleep disturbances suggests that diminished grip strength may serve as a risk factor for sleep disturbances among older adults [43,44]. These studies suggest that stronger grip strength is linked to a lower risk of sleep disturbances, and the study supports this negative association. One possible mechanism linking grip strength to sleep disturbances is the role of muscle mass and function in overall physical health. Sarcopenia, characterized by a decline in skeletal muscle mass and function, has been shown to have a significant association with sleep disturbances. Research on U.S. adults identified a positive association between sarcopenia and sleep disturbances, indicating that optimal muscle mass maintenance might positively influence sleep-related problems [45]. This indicates that interventions aimed at improving muscle strength is associated with sleep disturbances. Another potential mechanism is the bidirectional relationship between sleep disturbances and grip strength. A Mendelian randomization study identified causal links between sleep duration, sleep-wake disorders, and reduced grip strength. Found that shorter sleep duration correlated with reduced grip strength, whereas sleep-wake disorders were linked to decreased grip strength [21]. This suggests that sleep traits can directly influence muscle function, and vice versa, highlighting the importance of addressing sleep issues to maintain muscle health. Furthermore, the interaction between sleep duration

and grip strength has been explored in various populations. For instance, research involving Chinese middle-aged and older adults demonstrated that high grip strength could mitigate the association between short sleep duration and an increased risk of depressive symptoms [46]. This finding underscores the potential moderating role of grip strength in the relationship between sleep duration and mental health outcomes, suggesting that enhancing muscle strength could be a strategy to improve sleep quality.

### 4.3. Handgrip strength as a mediator between anemia and sleep disturbance

This study identified handgrip strength as a mediator between anemia and sleep disturbance. This discovery contributes novel insights to the existing body of literature, suggesting that anemia may adversely affect sleep quality. Anemia, defined by a deficiency in the quantity or quality of red blood cells, can impair oxygen delivery to tissues, potentially impacting muscle function and strength, which may manifest as reduced handgrip strength [47]. This reduction in handgrip strength could serve as a contributing factor to sleep disturbances in older adults [48]. Grip strength may mediate the relationship between anemia and sleep disturbances through several physiological pathways. The main hypothetical mediating pathway is muscle fatigue and recovery. Grip strength reflects overall muscle function. We hypothesize that anemia-induced muscle hypoxia, compounded by frequent comorbidities like micronutrient deficiencies and systemic inflammation, may contribute to daytime fatigue. However, whether this necessitates enhanced sleep-related recovery (potentially disrupting sleep architecture) remains speculative. While environmental hypoxia exacerbates inactivity-related muscle wasting [49], the distinct pathophysiology of anemia-related hypoxia requires direct investigation to establish such sleep-specific mechanisms.

### 4.4. Alternative explanations and potential confounding

The relationship between anemia and sleep disorders mediated by grip strength may involve alternative explanations and confounding factors. One potential mechanism is the inflammatory pathway, which could act as a common cause or confounder. Anemia has been associated with elevated pro-inflammatory cytokines, contributing to muscle depletion and reduced grip strength, thereby worsening sleep disturbances [50,51]. Notably, studies report a significant negative correlation between anemia and grip strength, particularly in men [51]. Moreover, the improvement of anemia can alleviate sleep-related breathing disorders by increasing hemoglobin levels, thereby improving daytime sleepiness [52]. Another plausible pathway involves energy metabolism, which may represent a confounding pathway or the direct effect of anemia. Impaired mitochondrial function due to anemia can disrupt adenosine triphosphate (ATP) production, leading to decreased grip strength and poorer sleep quality. Studies have indicated that there is a correlation between sleep quality and oxidative stress and inflammation, which may influence grip strength and sleep by affecting energy metabolism [53].

### 4.5. Limitation and strengths

This study exhibits several significant strengths. First, the CHARLS is a national study characterized by a large sample size, and its national representativeness has been widely recognized and acknowledged. Second, sociodemographic factors and covariates in this study were meticulously adjusted to minimize potential confounding effects. Third, the findings of this nationwide population-based study provide evidence not only for the prevention of sleep disturbances among older Chinese individuals but also serve as a reference for future research in other countries, particularly those that are developing. Nonetheless, this study has certain limitations. First, due to data limitations,the study did not evaluate whether anemia or the hemoglobin evel was associated with age, sex, etc examine the impact of anemia severity and type on the relationship between sleep disturbance, which may compromise the precision of the conclusions drawn. Second, potential bias might occur by using self-reported sleep disturbance as a substitute for objective measures of sleep disturbance. Third, the study employed cross-sectional data, which precludes establishing causal relationships between anemia,

handgrip strength, and sleep disturbances in the older adults. Evidence indicates that cross-sectional mediation analyses can produce significantly biased estimates of longitudinal parameters, particularly in cases of complete mediation [54,55]. Therefore, future studies should consider utilizing longitudinal data for mediation analysis. Fourth, due to the lack of information, a large proportion of the participants were excluded. If there is a difference between the included group and the excluded group, it may lead to selection bias. Fifth, although we accounted for most confounders, there might still be some unknown or unmeasured residual confounding. Finally, For the analysis, a substantial sample with national representativeness for individuals aged over 60 was utilized, thereby limiting the applicability of the results to this specific age group. Given the considerable sample size and minimal estimation deviation, survey weights were not incorporated into the analysis.

## 5. Conclusions

In conclusion, this cross-sectional study using CHARLS data highlights the strong association between anemia and sleep disturbances, handgrip strength may mediate this relationship. This study provides valuable references for improving the sleep quality of older adults.

## Supporting information

**S1 Table. Collinearity analysis result.**
(DOC)

**S2 Table. Characteristics of the study participants.**
(DOC)

**S3 Table. Associations of anemia and with handgrip strength.**
(DOC)

**S4 Table. Subgroup analysis of the association between anemia and sleep disturbance.**
(DOC)

**S5 Table. Subgroup analysis of the association between handgrip strength and sleep disturbance.**
(DOC)

**S1 Fig. Conceptual framework.**
(TIF)

**S2 Fig. Nonlinear associations of handgrip strength and sleep** disturbance. The median of the handgrip strength was used as the reference point. Solid and dashed lines represent the predicted value and 95% CI,respectively. Orange bars represent the distribution of the entire cohort. Adjusted for age, sex, educational level, marital status, residence, smoking status, drinking status, BMI, sleep duration, daytime napping duration, and 14 chronic conditions, Only 99% of the data is displayed.
(TIF)

**S1 Data. Raw data.**
(CSV)

## Acknowledgments

We express our gratitude to the China Center for Economic Research and the National School of Development at Peking University for supplying the data.

## Author contributions

**Conceptualization:** Xiaojiang Zhao.

**Data curation:** Jie Li, Zunyi Ma.

**Formal analysis:** Jie Li.

**Funding acquisition:** Xiaojiang Zhao.

**Investigation:** Jie Li.

**Resources:** Jie Li, Zunyi Ma.

**Software:** Zunyi Ma, Xiaojiang Zhao.

**Supervision:** Zunyi Ma.

**Visualization:** Jie Li.

**Writing – original draft:** Jie Li, Zunyi Ma, Xiaojiang Zhao.

**Writing – review & editing:** Xiaojiang Zhao.

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
