## [Decision Letter · Decision Letter 0]

24 Apr 2025

Dear Dr. Zhao,

Thank you for submitting your manuscript to PLOS ONE. After careful consideration, we feel that it has merit but does not fully meet PLOS ONE’s publication criteria as it currently stands. Therefore, we invite you to submit a revised version of the manuscript that addresses the points raised during the review process.

We look forward to receiving your revised manuscript.

Kind regards,

Qian Wu

Academic Editor

PLOS ONE

Journal Requirements:

“This work was supported by the Philosophy and Social Sciences Foundation of the Anhui Higher Education Institutions of China (2024AH052821 and 2024AH0052823). “

Reviewers' comments:

Reviewer's Responses to Questions

**Comments to the Author**

1. Is the manuscript technically sound, and do the data support the conclusions?

Reviewer #1: Partly

Reviewer #2: Partly

Reviewer #3: Partly

Reviewer #4: Partly

Reviewer #5: Yes

2. Has the statistical analysis been performed appropriately and rigorously?

Reviewer #1: Yes

Reviewer #2: I Don't Know

Reviewer #3: Yes

Reviewer #4: Yes

Reviewer #5: Yes

3. Have the authors made all data underlying the findings in their manuscript fully available?

Reviewer #1: Yes

Reviewer #2: Yes

Reviewer #3: No

Reviewer #4: Yes

Reviewer #5: Yes

4. Is the manuscript presented in an intelligible fashion and written in standard English?

Reviewer #1: Yes

Reviewer #2: Yes

Reviewer #3: No

Reviewer #4: Yes

Reviewer #5: Yes

Reviewer #1: Reviewer Summary and Comments

Overall Assessment:

This manuscript explores the mediating role of handgrip strength in the association between anemia and sleep disturbance among older Chinese adults, using data from the 2015 CHARLS cohort. The paper is written in clear and fluent English, with a well-structured and logically developed narrative. The research question is timely and of potential public health relevance. However, several conceptual and methodological issues limit the interpretability and robustness of the findings. Below, I provide detailed comments by section.

Abstract & Introduction:

The hypothesis is clearly stated, but the rationale for handgrip strength as a mediator rather than a moderator or independent predictor is insufficiently justified.

I recommend briefly outlining the plausible biological mechanisms linking anemia, handgrip strength, and sleep disturbance more clearly in the introduction.

Section 2.2 – Assessment of Key Variables:

Sleep disturbance is assessed using a single self-reported item ("My sleep was restless"), which limits validity and depth. This should be acknowledged as a limitation.

I recommend the authors suggest the use of objective, multidimensional sleep-tracking tools (e.g., actigraphy, wearable devices such as Apple Watch or Oura Ring) in future studies to enhance measurement precision and reduce bias.

The definition of anemia follows WHO cutoffs but does not account for age-related changes in hemoglobin, potentially leading to misclassification. This should be addressed or at least mentioned.

Section 3.1 – Descriptive Results:

The binary classification of sleep disturbance is simplistic and may obscure meaningful gradations. Consider justifying this dichotomization more clearly or exploring alternative modeling strategies.

Section 3.2 – Regression Analysis (Table 2):

The regression models are statistically sound and robust across adjustments. However, effect sizes are modest (e.g., OR = 1.17 for anemia, OR = 0.98 for handgrip strength).

Authors should discuss the clinical significance of these findings.

The restricted cubic spline (RCS) curve (Supplementary Figure) appears U-shaped rather than linear. Although the test for non-linearity was non-significant (p = 0.88), the visual pattern should be acknowledged. In particular, lower and higher handgrip strength values both seem to be associated with reduced odds of sleep disturbance, with a peak in odds ratio around 26.5 kg. This contradicts the narrative of a simple inverse relationship and should be interpreted with caution.

Section 3.3 – Correlations and Mediation Analysis:

The rationale for the mediation model (anemia → grip strength → sleep disturbance) remains conceptually weak. The manuscript lacks a physiological explanation of how daytime grip strength causally affects nighttime sleep quality.

Mediation requires an explanatory pathway, but this is not well substantiated. For example, muscle inactivity during sleep challenges the premise.

Correlations are statistically significant but weak (e.g., r = 0.04). The indirect effect accounts for only 15.67% of the total effect. These limitations should be discussed.

Also, two sections are labeled 3.3. This numbering issue should be corrected.

Section 3.4 – Subgroup Analyses (Supplementary Tables 1 & 2):

Subgroup analyses do not reveal any significant interactions. Several strata are underpowered, with wide confidence intervals.

The authors should clarify whether these analyses were hypothesis-driven or exploratory, and explain their relevance.

Supplementary Table 1 (labeled Table 3) shows no significant interaction between anemia and sleep disturbance across subgroups. The observed ORs are modest and variable, with wide confidence intervals in several underpowered strata, limiting interpretability.

The same applies to “Supplementary Table 2” (labeled Table 4) regarding subgroup analyses of handgrip strength and sleep disturbance — the manuscript would benefit from clearer table referencing and justification of these additional analyses.

Discussion:

The discussion overstates the strength and causality of findings. Phrases such as "handgrip strength mediates..." should be tempered to "may mediate..." or "suggests a potential mediating role."

The narrative includes well-referenced physiological pathways (iron, sarcopenia, sleep-wake regulation), which help contextualize the results. However, this should be clearly separated from what the data actually support.

The rationale for mediation remains weak: as muscles are at rest during sleep, the causal pathway from grip strength to sleep remains speculative.

Limitations & Conclusion:

Key limitations are acknowledged, including cross-sectional design, lack of objective sleep measures, and absence of anemia severity stratification.

However, the conclusion presents intervention strategies as if causality had been established. This should be revised to reflect the exploratory and correlational nature of the study.

Recommendation:

Major Revision.

The study is relevant and methodologically competent, but the theoretical justification for the mediation model needs significant improvement. Claims of mediation and causal direction should be rephrased cautiously, and several methodological limitations should be better addressed. With these revisions, the manuscript could make a valuable contribution to the literature.

Reviewer #2: The present study investigates the relationship between anemia and sleep disturbances among older adults in China: The mediating role of handgrip strength. The present study investigates the mediating role of handgrip strength in the relationship between anemia and sleep disturbances. However, some comments have been found while reading the paper which need to be revised. The following comments are described as given:

1. The quantitative results beside the qualitative results should be added in the abstract should.

2. The structure of the paper should be clearly described at the end of introduction section.

3. Literature review should be separate from the introduction since the novelty is not clear.

4. What is the research gap between the present study and studies in the literature review? Authors should highlight this importance at the end of literature review. What is the novelty of the present work in comparison with other works?

5. In order to extend the domain of the present study in comparison with other studies and increase the contribution of the present paper with other evaluation methods in anemia and sleep disturbances among older adults and other fields, the following references could be added which are:

• Orexin Neurons to Sublaterodorsal Tegmental Nucleus Pathway Prevents Sleep Onset REM Sleep-Like Behavior by Relieving the REM Sleep Pressure. Research, 7, 0355. doi: 10.34133/research.0355

• The relationship between handgrip strength and cognitive function among older adults in China: Functional limitation plays a mediating role. Journal of Affective Disorders, 347, 144-149.

• Nip it in the bud: the impact of China’s large-scale free physical examination program on health care expenditures for elderly people. Humanities and Social Sciences Communications, 12(1), 27. doi: 10.1057/s41599-024-04295-5

• Factors associated with low handgrip strength in older people: data of the Study of Chronic Diseases (Edoc-I). BMC Public Health, 20, 1-10.

6. Authors should add a section regarding the data description, initial data statistics, sample stability testing. How did authors understand that the sample data is enough? Has the survey response stability been implemented? Please add a section regarding this importance.

7. In the research method, what statistical tests have been conducted to evaluate the normality or abnormality of data? Wasn’t it better to use some statistical tests for evaluating normality or abnormality of data and then use Pearson or Spearman based on the data for investigating the correlation?

8. Why multivariable logistic regression analysis was used? How about simple regression or multi regression models? Why authors didn’t use the mentioned models? How did they decide to use multivariable logistic regression analysis?

9. Conclusion part should be extended based on the whole paper since it is the main part of the present work.

Reviewer #3: This study offers a valuable contribution by examining the relationship between anemia and sleep disturbances among older adults in China, particularly relevant in the current context where many countries are transitioning into aging societies. The findings are of particular importance given the implications of poor sleep quality for overall well-being and life satisfaction among the elderly. Notably, the study also explores the mediating role of handgrip strength, a novel angle that may help to inform strategies aimed at improving sleep quality.

However, I would like to point out several areas where further development and refinement are warranted

General Comments

• Manuscript structure: The writing throughout the manuscript would benefit from refinement for clarity and academic tone. Redundant expressions (e.g., "association relationship") should be revised for conciseness and precision.

• References: The citation style needs to be standardized according to journal guidelines. Additionally, the full text or supplemental material should be provided for complete data transparency.

Section-Specific Comments

1. Abstract

• The objective section could be more clearly articulated, and redundant phrasing should be avoided. For example, the sentences: “This study seeks to examine the mediating role of handgrip strength in the relationship between anemia and sleep disturbances.”

and“Additionally, bootstrap tests were performed to assess the mediating role of handgrip strength in the association between anemia and sleep disturbances.”

essentially convey the same point and should be consolidated.

• In the results section, consider replacing the word "correlation" if the results are based on odds ratios (OR), which imply associative or causal inference rather than simple correlation.

2. Introduction

• The citation to Zhong et al., 2024, which states a relationship between grip strength and sleep disturbances, appears to be inaccurate. Based on the referenced study, Zhong et al. examined waist circumference and sleep disorders—not grip strength. This needs to be corrected.

• The rationale for proposing handgrip strength as a mediator is currently weak. A theoretical or empirical basis should be introduced to support its mediating role—such as references to prior studies or physiological mechanisms that connect muscle strength, anemia, and sleep disturbances.

• The statement “several studies” is misleading when only one source (e.g., Denny et al., 2006) is cited. Multiple citations (≥2) are expected to support such a claim.

• Vague terms like "complexity" should be elaborated upon or supported with concrete examples or sources—what constitutes the "complexity" of sleep disturbances?

3. Methods

• The rationale behind using the CHARLS questionnaire to assess sleep disturbances should be justified. Does this single-question format reliably capture sleep quality among older adults? Consider comparing it with validated tools like the Pittsburgh Sleep Quality Index (PSQI).

• Details on how handgrip strength was measured are limited. Was the method aligned with WHO, NIH, or other standardized protocols? Including a reference or illustration of the assessment method would enhance clarity.

4. Results

• Table 1: While p-values are presented, the specific statistical tests used (e.g., chi-square, t-test) are not indicated. In addition, the standard deviations for handgrip strength vary considerably between groups. Reporting an effect size (e.g., Cohen’s d) would enhance interpretability.

• Table 2: The odds ratio for handgrip strength should specify the unit, such as “per 1 kg increase” or “per standard deviation increase,” to clarify interpretation.

• Table 3: Although correlations may be statistically significant, the magnitude is very weak and should be acknowledged as such.

• Figure 2 should be revised to include additional methodological details, such as the statistical technique used (e.g., bootstrapping), and the confidence intervals (CIs) for the indirect effect, if available. Including this information would enhance the completeness and interpretability of the mediation analysis.

Reviewer #4: Thank you for this important work. I enjoyed reviewing your work. The major strength of this manuscript were the flow your writing and the methodology aspects including a longitudinal dataset. However, I have some critical points for you to address in the attached document. If you find some of my comments not relevant, please feel free to respond accordingly. I wish you and your coauthors the best, and I hope that my comments will help enhance the quality of your work.

Reviewer #5: The manuscript investigates the mediating role of handgrip strength in the association between anemia and sleep disturbances among older adults in China by using CHARLS 2015 data. However, authors should clarify:

1.Why iron status was not available and how its absence might limit interpretation.

2. How multicollinearity was assessed among anemia, grip strength, chronic conditions.

3. Whether the logistic regression models included survey weights or clustering corrections given complex sampling design.

4. Whether interaction terms in subgroup analyses were statistically tested, if so, include p-values for interaction.

5. The methodological details such as variable coding, weighting, handling of missing data.

The correlation coefficients in Table 3 are statistically significant but weak in magnitude like r = 0.04 so one should be careful in interpreting practical relevance.

Provide more interpretative commentary in the main text for Figures. Also, tables may be placed in main text.

Discuss whether sleep disturbance assessment via a single-item self-report question might underestimate sleep problems.

The interpretation of handgrip strength should be presented with caution and may be tested with longitudinal data in future research.

**Do you want your identity to be public for this peer review?** For information about this choice, including consent withdrawal, please see our Privacy Policy

Reviewer #1: **Yes: ** Thomas C. Carmine MD

Reviewer #2: No

Reviewer #3: No

Reviewer #4: **Yes: ** Aman Shrestha

Reviewer #5: No

---

## [Author Response · Author response to Decision Letter 1]

17 May 2025

Response to Editor

Dear Dr. Qian Wu

Thank you very much for your letter and the comments about our paper “The Relationship between anemia and sleep disturbances among Older Adults in China: The Mediating Role of handgrip strength (Manuscript ID: 517084bef51b020d)”. After carefully studying the reviewers' comments and advice, we have made corresponding changes to the paper. Revised sections are identified with red text in the paper.

There are no conflicts of interest regarding this work. All authors have read the revised manuscript and approved its submission to the PLOS ONE. Please do not hesitate to contact us if we can be of any further assistance.

Thank you and best regards.

Yours Sincerely

Xiaojiang Zhao

mail: zhaoxiaojiang2010@163.com

Response to Editor

Question 1: Please ensure that your manuscript meets PLOS ONE's style requirements, including those for file naming. The PLOS ONE style templates can be found at https://journals.plos.org/plosone/s/file?id=wjVg/PLOSOne_formatting_sample_main_body.pdf and https://journals.plos.org/plosone/s/file?id=ba62/PLOSOne_formatting_sample_title_authors_affiliations.pdf.

Answer: Thank you for your valuable suggestions on the manuscript. We have carefully checked and ensured that the manuscript fully complies with the format requirements of PLOS ONE, including the file naming conventions.

Question 2: We note that the grant information you provided in the ‘Funding Information’ and ‘Financial Disclosure’ sections do not match. When you resubmit, please ensure that you provide the correct grant numbers for the awards you received for your study in the ‘Funding Information’ section.

Answer: Thank you for your careful review of our manuscript and for bringing this important discrepancy to our attention. We sincerely apologize for the oversight in the funding information consistency between sections. We have now carefully verified our records and made the following corrections to ensure complete alignment: Funding Information section:[Updated grant details with correct grant numbers, e.g., "This work was supported by the Philosophy and Social Sciences Foundation of the Anhui Higher Education Institutions of China (2024AH052821) and Philosophy and Social Sciences Foundation of the Anhui Higher Education Institutions of China (2024AH0052823)" ]( lines 307-309). We appreciate your vigilance in helping us maintain the highest standards of accuracy in our publication. Please let us know if any additional clarifications are needed.

Question 3: Thank you for stating the following financial disclosure:

“This work was supported by the Philosophy and Social Sciences Foundation of the Anhui Higher Education Institutions of China (2024AH052821 and 2024AH0052823).“Please state what role the funders took in the study. If the funders had no role, please state: "The funders had no role in study design, data collection and analysis, decision to publish, or preparation of the manuscript."If this statement is not correct you must amend it as needed. Please include this amended Role of Funder statement in your cover letter; we will change the online submission form on your behalf.

Answer: Thank you for your comment regarding the funding disclosure. The funders (Philosophy and Social Sciences Foundation of the Anhui Higher Education Institutions of China) had no role in study design, data collection and analysis, decision to publish, or preparation of the manuscript. Please find the following statement included in our cover letter: "The funders had no role in study design, data collection and analysis, decision to publish, or preparation of the manuscript." We kindly request the editorial office to update the online submission form accordingly on our behalf. Please let us know if any additional clarification is needed.

Question 4: Please include captions for your Supporting Information files at the end of your manuscript, and update any in-text citations to match accordingly. Please see our Supporting Information guidelines for more information: http://journals.plos.org/plosone/s/supporting-information.

Answer: Thank you for your valuable feedback. We sincerely appreciate your time and effort in reviewing our manuscript. In response to your comment regarding Supporting Information captions and citations, we have: Added detailed captions for all Supporting Information files at the end of the manuscript, following PLOS ONE's guidelines( lines 290-301).

Response to Reviewers

Dear Reviewers.

Thank you very much for your letter and the comments about our paper “The Relationship between anemia and sleep disturbances among Older Adults in China: The Mediating Role of handgrip strength (Manuscript ID: 517084bef51b020d)”. After carefully studying the reviewers' comments and advice, we have made corresponding changes to the paper. Revised sections are identified with red text in the paper.

There are no conflicts of interest regarding this work. All authors have read the revised manuscript and approved its submission to the PLOS ONE. Please do not hesitate to contact us if we can be of any further assistance.

Thank you and best regards.

Yours Sincerely

Xiaojiang Zhao

mail: zhaoxiaojiang2010@163.com

Reviewer #1:

Reviewer Summary and Comments Overall Assessment: This manuscript explores the mediating role of handgrip strength in the association between anemia and sleep disturbance among older Chinese adults, using data from the 2015 CHARLS cohort. The paper is written in clear and fluent English, with a well-structured and logically developed narrative. The research question is timely and of potential public health relevance. However, several conceptual and methodological issues limit the interpretability and robustness of the findings. Below, I provide detailed comments by section.

Question 1: Abstract & Introduction: The hypothesis is clearly stated, but the rationale for handgrip strength as a mediator rather than a moderator or independent predictor is insufficiently justified. I recommend briefly outlining the plausible biological mechanisms linking anemia, handgrip strength, and sleep disturbance more clearly in the introduction.

Answer: Thank you for your constructive suggestions. Following the your suggestion In the introduction, we have added the description of the biological mechanism that links anemia, handgrip strength, and sleep disturbance( lines 66-71).

Question 2: section 2.2 – Assessment of Key Variables: Sleep disturbance is assessed using a single self-reported item ("My sleep was restless"), which limits validity and depth. This should be acknowledged as a limitation. I recommend the authors suggest the use of objective, multidimensional sleep-tracking tools (e.g., actigraphy, wearable devices such as Apple Watch or Oura Ring) in future studies to enhance measurement precision and reduce bias. The definition of anemia follows WHO cutoffs but does not account for age-related changes in hemoglobin, potentially leading to misclassification. This should be addressed or at least mentioned.

Answer: Thank you for your valuable suggestion. As a response to "The limitations related to assessment of sleep disturbance and anemia"in response, We have revised the relevant content in the discussion section (Line 276-277).

Question 3: Section 3.1 – Descriptive Results: The binary classification of sleep disturbance is simplistic and may obscure meaningful gradations. Consider justifying this dichotomization more clearly or exploring alternative modeling strategies.

Answer: Thank you again for your positive comments and valuable suggestions. Based on previous studies(PMID: 39670150, PMID: 31332159, and PMID: 40055626 ), we have added a description of binary classification of sleep disturbances in the methods section( lines 100-104).

Question 4: Section 3.2 – Regression Analysis (Table 2): The regression models are statistically sound and robust across adjustments. However, effect sizes are modest (e.g., OR = 1.17 for anemia, OR = 0.98 for handgrip strength). Authors should discuss the clinical significance of these findings. The restricted cubic spline (RCS) curve (Supplementary Figure) appears U-shaped rather than linear. Although the test for non-linearity was non-significant (p = 0.88), the visual pattern should be acknowledged. In particular, lower and higher handgrip strength values both seem to be associated with reduced odds of sleep disturbance, with a peak in odds ratio around 26.5 kg. This contradicts the narrative of a simple inverse relationship and should be interpreted with caution.

Answer:

Thank you for your suggestions, they are excellent. As a response to "Regression Analysis (Table 2)"in response, thank you again for your recognition of the research methods and statistical robustness. We fully agree that the discussion of the clinical significance of the effect size is very important. Relevant content has now been added to the discussion section( lines 260-264). As a response to "The restricted cubic spline (RCS) curve (Supplementary Figure1)"in response, In our study, Supplementary Figure 1 uses arestricted cubic spline curve (RCS) to investigate the dose-response relationship between handgrip strength and sleep disturbance, adjusting for confounders according to the logistic regression model 3, which showed a non-linear relationship between handgrip strength and sleep disturbance. However, the reference point 26.5 kg in the graph represents the median handgrip strength (Supplementary Figure 1). To address such misunderstandings, we added "The median of the handgrip strength was used as the reference point" in the legend( lines 512).

Question 5: Section 3.3 – Correlations and Mediation Analysis: The rationale for the mediation model (anemia → grip strength → sleep disturbance) remains conceptually weak. The manuscript lacks a physiological explanation of how daytime grip strength causally affects nighttime sleep quality. Mediation requires an explanatory pathway, but this is not well substantiated. For example, muscle inactivity during sleep challenges the premise. Correlations are statistically significant but weak (e.g., r = 0.04). The indirect effect accounts for only 15.67% of the total effect. These limitations should be discussed. Also, two sections are labeled

Answer: Thank you for raising these important points. We appreciate the opportunity to clarify and strengthen our rationale. Below is our point-by-point response: As a response to "Explanation of the physiological pathway of mediation model"in response, we acknowledge the need for clearer physiological justification and have expanded our discussion( lines 254-261). As a response to "Correlation and mediating effect"in response, We have added the relevant content in the discussion section (Lines 261-265). As a response to "Labeling error"in response, thank you for carefully reviewing our manuscript and pointing out this numbering error. We have conducted a comprehensive review of the chapter numbers and made the following corrections. Thank you again for helping us improve the standardization of the thesis. If there are any other adjustments that need to be made, please inform us at any time.

Question 6: Section 3.4 – Subgroup Analyses (Supplementary Tables 1 & 2): Subgroup analyses do not reveal any significant interactions. Several strata are underpowered, with wide confidence intervals. The authors should clarify whether these analyses were hypothesis-driven or exploratory, and explain their relevance. Supplementary Table 1 (labeled Table 3) shows no significant interaction between anemia and sleep disturbance across subgroups. The observed ORs are modest and variable, with wide confidence intervals in several underpowered strata, limiting interpretability. The same applies to “Supplementary Table 2” (labeled Table 4) regarding subgroup analyses of handgrip strength and sleep disturbance — the manuscript would benefit from clearer table referencing and justification of these additional analyses.

Answer:

Thank you for your thoughtful comments regarding our subgroup analyses. We appreciate the opportunity to clarify these important points. Regarding the nature of subgroup analyses, we acknowledge that our presentation could have been clearer. These analyses were primarily exploratory, intended to examine whether the observed associations between anemia/handgrip strength and sleep disturbance might differ across population subgroups. We have refined the relevant content in the Subgroup Analyses section(Lines 184-199). Additionally, regarding the annotation errors in the supplementary table, we have made corrections(S4 Table and S5 Table).

Question 7: Discussion: The discussion overstates the strength and causality of findings. Phrases such as "handgrip strength mediates..." should be tempered to "may mediate..." or "suggests a potential mediating role." The narrative includes well-referenced physiological pathways (iron, sarcopenia, sleep-wake regulation), which help contextualize the results. However, this should be clearly separated from what the data actually support. The rationale for mediation remains weak: as muscles are at rest during sleep, the causal pathway from grip strength to sleep remains speculative.

Answer: Thank you for your suggestions, we fully agree. Below is our point-by-point response: in response to the expression "the term 'mediates' is inappropriate," we sincerely appreciate the reviewer's insightful comment regarding the need for more cautious language when describing the mediation findings. As suggested, we have revised all definitive causal claims (e.g., 'mediates' → 'may mediate') throughout the Discussion to better reflect the observational nature of our data (Lines 201, 204, 254). As a response to "mediation mechanism"in response, We have added the relevant content in the discussion section (Lines 254-260).

Question 6: Limitations & Conclusion: Key limitations are acknowledged, including cross-sectional design, lack of objective sleep measures, and absence of anemia severity stratification. However, the conclusion presents intervention strategies as if causality had been established. This should be revised to reflect the exploratory and correlational nature of the study.

Answer: Thank you for your suggestions, we sincerely thank the reviewer for highlighting the need to align our conclusions with the study's correlational design. We have revised the Conclusion section to explicitly emphasize the exploratory nature of our findings and removed any language implying causal recommendations (Line 277-280).

Question 7: Recommendation: Major Revision. The study is relevant and methodologically competent, but the theoretical justification for the mediation model needs significant improvement. Claims of mediation and causal direction should be rephrased cautiously, and several methodological limitations should be better addressed. With these revisions, the manuscript could make a valuable contribution to the literature.

Answer: Thank you again for your valuable opinions on this article. The insufficiency of the theoretical framework and the problem of causal inference you pointed out are indeed crucial to the quality of the article. We have made the modifications as per your suggestions (see the modification section above).

Reviewer #2:

The present study investigates the relationship between anemia and sleep disturbances among older adults in China: The mediating role of handgrip strength. The present study investigates the mediating role of handgrip strength in the relationship between anemia and sleep disturbances. However, some comments have been found while reading the paper which need to be revised.The following comments are described as given:

Question 1: The quantitative results beside the qualitative results should be added in the abstract should.

Answer:

We sincerely appreciate your constructive suggestion. According to the suggestions, we have added the key quantitative findings in the abstract(Lines 28-30).

Question 2: The structure of the paper should be clearly described at the end of introduction section.

A

---

## [Decision Letter · Decision Letter 1]

12 Jun 2025

Dear Dr. Zhao,

Thank you for submitting your manuscript to PLOS ONE. After careful consideration, we feel that it has merit but does not fully meet PLOS ONE’s publication criteria as it currently stands. Therefore, we invite you to submit a revised version of the manuscript that addresses the points raised during the review process.

Please make peer-to-peer modifications to the reviewer's comments.

We look forward to receiving your revised manuscript.

Kind regards,

Qian Wu

Academic Editor

PLOS ONE

Journal Requirements:

Reviewers' comments:

Reviewer's Responses to Questions

**Comments to the Author**

Reviewer #1: All comments have been addressed

Reviewer #2: All comments have been addressed

Reviewer #4: (No Response)

Reviewer #5: All comments have been addressed

2. Is the manuscript technically sound, and do the data support the conclusions?

Reviewer #1: Partly

Reviewer #2: Yes

Reviewer #4: Yes

Reviewer #5: Yes

3. Has the statistical analysis been performed appropriately and rigorously?

Reviewer #1: Yes

Reviewer #2: Yes

Reviewer #4: Yes

Reviewer #5: Yes

4. Have the authors made all data underlying the findings in their manuscript fully available?

Reviewer #1: Yes

Reviewer #2: Yes

Reviewer #4: Yes

Reviewer #5: Yes

5. Is the manuscript presented in an intelligible fashion and written in standard English?

Reviewer #1: Yes

Reviewer #2: Yes

Reviewer #4: Yes

Reviewer #5: Yes

Reviewer #1: Dear Authors,

Thank you again for your diligent revisions. Your softened causal language, expanded biological discussion, and clearer hypothesis framing have greatly improved the manuscript. I recommend acceptance pending a few minor edits and clarifications, especially around distinguishing mediation from confounding and correcting a mis-applied citation. Please address the points below—organized by line number:

1. Line 51–52

Issue: Wording implies anemia is a subtype of “nutritional deficiencies,” and Ref. 15 does not address anemia.

Fix: Either remove “anemia” from this clause or replace it with a citation specifically linking anemia to sleep disturbances.

2. Lines 106–108

Issue: WHO hemoglobin cutoffs (<12.0 g/dL women; <13.0 g/dL men) omit age-related declines.

Fix: Add a brief note in Methods acknowledging this limitation (and, if feasible, mention a sensitivity analysis with age-specific ranges).

3. Lines 203–205

Issue: Typo “sleep disturbanced breathing.”

Fix: Correct to “sleep-disordered breathing.”

4. Lines 206–207

Issue: “Lower anemia” is imprecise.

Fix: Change to “lower hemoglobin levels” or “milder anemia.”

5. Lines 195–199 & 241–248

Issue: Some definitive causal verbs remain (“mediates,” “could alleviate sleep disturbances”).

Fix: Replace with circumspect phrasing (“may mediate,” “is associated with,” “could reflect”).

6. Numbering and Cross-References

Issue: Section/table labels (e.g., S4 vs. S5) and in-text citations sometimes mismatch.

Fix: Ensure every cited figure/table corresponds to the correct label.

7. Clarify Mediation vs. Confounding

Several passages conflate mediator pathways with confounding influences. Please revise as follows:

Iron-Deficiency Mechanism (Lines 206–217):

Label body iron status as a confounder, not a mediator, or move to Limitations, noting ferritin wasn’t measured.

Inflammatory Pathways (Lines 248–252):

Recast inflammation as a shared cause of both reduced grip strength and poor sleep, or explicitly justify how it transmits effects through grip strength.

Bidirectional Interplay (Lines 246–249):

Choose one primary direction (grip → sleep) for mediation, and present the reverse (sleep → grip) as an alternative hypothesis or effect modifier—not part of the same mediation pathway.

8. Muscle Fatigue “Recovery” Mechanism (Lines 249–252)

Current text:

“Grip strength reflects overall muscle function, and the anemia-induced weakness can cause increased daytime fatigue, which may necessitate more intense recovery during sleep and potentially disrupt sleep patterns[47].”

Issues:

The cited Debevec et al. study examines normobaric hypoxia and bed-rest muscle wasting—environmental oxygen deprivation under inactivity—not anemia-induced hypoxia or sleep architecture.

Anemia-related hypoxia co-occurs with other factors (e.g., micronutrient deficiencies, inflammation), making it not equivalent to pure environmental hypoxia.

Fix: Either

Provide a sleep-specific citation that directly links anemia-induced muscle fatigue or elevated metabolic demand to disrupted sleep stages, or

Flag as speculative, for example:

“This remains hypothetical—while environmental hypoxia can exacerbate inactivity-related muscle wasting[Debevec et al. 2018], anemia-related hypoxia (often accompanied by micronutrient deficiencies and inflammation) may not have the same effects on sleep architecture, and this pathway requires direct study.”

Once these edits and clarifications are made, the manuscript will present a coherent, well-justified narrative—appropriately cautious about causality and clear in its distinction between mediators and confounders.

Thank you for your dedication to improving the clarity and rigor of this important work. I look forward to reviewing the final version.

Best regards,

Reviewer 1

Reviewer #2: The authors addressed and did the revision. So, the paper can be accepted and published based on suitable responses to questions and revisions.

Reviewer #4: Thank you for sincerely addressing and incorporating my previous comments and recommendations. I hope they were helpful in enhancing the quality of your work and promoting the rigor of your research.

I do, however, have a few remaining suggestions that were not fully addressed in your previous response. While my training in Epidemiology typically advises against using cross-sectional data for mediation analysis, I understand that such approaches are commonly employed in the literature to explore associations. Given this context, I would recommend that you explicitly and critically acknowledge this methodological limitation in your study.

Additionally, if your analysis was constrained to using only Wave 3 of the CHARLS dataset due to the availability of relevant variables, it would be helpful to clearly state this in your manuscript. Otherwise, it would be worth considering whether a longitudinal approach might strengthen the validity of your findings.

Please find my final comments below. And, thank you for this important piece of work.

Abstract

Methods

Since you do not entail what specific inclusion and exclusion criteria were applied in the abstract, you may just say that 6,057 is your analytic sample. It is by default understood that the analytic sample is a result of certain operation and treatments to the original dataset.

To save up some space in the abstract, you may only state the variable categories inside the brackets. For example, sleep (yes/no) and anemia (present/absent).

Please discuss with the expert in your team if binary logistic regression or multivariable logistic regression will be more appealing term since your dependent variable is binary and you only have one primary independent variable. However, since you are controlling more than one independent variable, multivariable logistic regression also fits. I am fine with any of the two terms.

Introduction

<format> Please check the PLOS One formatting guidelines if there should be space before [] sign used for in-text citation. If yes, make changes throughout the manuscript.

<1st paragraph> While I appreciate the operationalization of "sleep disturbance" in the Methods section, I would respectfully suggest that it might be beneficial to introduce and contextualize the term earlier in the manuscript based on existing literature, ideally in the Introduction. Providing a brief overview of how "sleep disturbance" is defined in the scientific literature—or acknowledging the lack of a standardized definition—could help readers better understand the scope and relevance of your study. Given that "sleep disturbance" can encompass a range of issues such as insomnia, obstructive sleep apnea, and excessive daytime sleepiness, clarifying what is meant in your study from the outset could enhance the clarity and impact of your work. The operational definition fits well under the methods section.

<4th paragraph> Apologies if I may have overlooked it, but I did not clearly see the theoretical foundation referenced in lines 75–78. If it is present, it may benefit from clearer articulation. Additionally, as a general convention, the theoretical framework is typically introduced before the last paragraph in the introduction section where you describe the research objective. Also, ideally the conceptual framework positioned after the research objectives and before the Methods section. Presenting it there can help anchor the study's rationale and guide readers through the analytical approach more effectively. I would recommend using conceptual framework than DAGS since your study is cross-sectional.

Methods

<1st paragraph> If CHARLES is a longitudinal study, not a part of longitudinal study, then please correct the statement where it means that CHARLES is a part of a longitudinal study.

<major> Thank you for your response to the following question. However, I would like to request you consider elaborating your response to answer all the questions below. This is one of the most important limitations of your study that you need to carefully address.

Given that the CHARLS dataset is longitudinal, your decision to rely solely on Wave 3 for a cross-sectional analysis warrants a clear and well-reasoned justification—particularly in light of your acknowledgment that research on the causal

relationship between anemia and sleep disturbance remains limited. What distinguishes your cross-sectional study from existing ones that have already explored this association? Does your work offer new insights, or does it largely replicate

previously established findings? Additionally, it is important to address the limitations inherent in using cross-sectional data to test mediating relationships. Such analyses are generally viewed as exploratory rather than confirmatory, due to the inability to establish temporal ordering and causal direction. Given you already have access to longitudinal data, why motivates you to prefer a cross-sectional analysis? Please, clarify the rationale behind focusing exclusively on a single wave for establishing the validity and contribution of your study.

Your response was:

Thank you for the important questions you raised about our research

methods. Regarding our decision to use CHARLS Wave 3 data for cross-sectional

analysis rather than longitudinal analysis, we understand your concerns and are willing

to explain our considerations in detail. First of all, our research builds on previous

literature on the association between anemia and sleep disturbances, but we focused

on older Chinese adults and explored the mediating role of handgrip strength in the

association between anemia and sleep disturbances. Secondly, the response rate and

data quality of Wave 3 are the most ideal among all waves of CHARLS. In the future,

we will follow your suggestion and conduct research using the complete longitudinal

data of CHARLS.

<comments response to your=">"

While mediation analysis can be conducted using cross-sectional data for exploratory purposes, it is ideally suited to longitudinal designs due to the need for temporal ordering to support causal inference. Given that CHARLS is a longitudinal dataset, it would be helpful to clarify whether the key variables were available across multiple waves. If not, this would justify the use of cross-sectional data. However, the rationale provided—based on response rate and data quality—does not fully address the choice to forgo a longitudinal approach. If you were only exploring the relationship without including the mediation, cross-sectional dataset would be fine. But, since you included mediation, I recommend explicitly acknowledging this limitation in the manuscript, especially considering the availability of the variables of interest in multiple waves of the longitudinal data.

Also, you might consider revisiting the rationale on focusing older adults, as I found several pre-existing observational studies exploring this relationship in Chinese adult and older adult population. I included some of them below. Your rationale should justify what different or important exploration your study is contributing in light of pre-existing literature on this topic. These are crucial to ensure rigor of the study.

Cheng C, Chen X, Zhang L, et al. A Risk Correlative Model for Sleep Disorders in Chinese Older Adults Based on Blood Micronutrient Levels: A Matched Case-Control Study. Nutrients. 2024;16(19):3306. Published 2024 Sep 29. doi:10.3390/nu16193306

Neumann SN, Li JJ, Yuan XD, et al. Anemia and insomnia: a cross-sectional study and meta-analysis. Chinese Medical Journal. 2021;134(6):675-681. doi:10.1097/CM9.0000000000001306

Kara O, Elibol T, Koc Okudur S, Smith L, Soysal P. Associations between anemia and insomnia or excessive daytime sleepiness in older adults. Acta Clinica Belgica. 2023;78(3):223-228. doi:10.1080/17843286.2022.2116895

Statistical Methods

Please add a citation which you refer to using VIF cut-off point of 5 or above.</comments></major></format>

Reviewer #5: The authors have shown exceptional responsiveness and a clear effort to implement substantive changes. The manuscript now:

Meets the standards of methodological rigor, clearly distinguishes correlation from causation, and demonstrates scientific and editorial maturity.

**Do you want your identity to be public for this peer review?** For information about this choice, including consent withdrawal, please see our Privacy Policy

Reviewer #1: No

Reviewer #2: No

Reviewer #4: **Yes: ** Aman Shrestha

Reviewer #5: No

---

## [Author Response · Author response to Decision Letter 2]

11 Jul 2025

Response to Editor

Dear Dr. Qian Wu

Thank you very much for your letter and the comments about our paper“The Relationship between anemia and sleep disturbances among Older Adults in China: The Mediating Role of handgrip strength (Manuscript ID: 517084bef51b020d)”. According to your suggestion, we have verified all cited references against PubMed databases and confirmed that none of the cited papers have been retracted.

After carefully studying the reviewers' comments and advice, we have made corresponding changes to the paper. Revised sections are identified with red text in the paper.

There are no conflicts of interest regarding this work. All authors have read the revised manuscript and approved its submission to the PLOS ONE. Please do not hesitate to contact us if we can be of any further assistance.

Thank you and best regards.

Yours Sincerely

Xiaojiang Zhao

E-mail: zhaoxiaojiang2010@163.com

Response to Reviewers

Dear Reviewers.

Thank you very much for your letter and the comments about our paper “The Relationship between anemia and sleep disturbances among Older Adults in China: The Mediating Role of handgrip strength (Manuscript ID: 517084bef51b020d)”. After carefully studying the reviewers' comments and advice, we have made corresponding changes to the paper. Revised sections are identified with red text in the paper.

There are no conflicts of interest regarding this work. All authors have read the revised manuscript and approved its submission to the PLOS ONE. Please do not hesitate to contact us if we can be of any further assistance.

Thank you and best regards.

Yours Sincerely

Xiaojiang Zhao

mail: zhaoxiaojiang2010@163.com

Response to Reviewer #1

We sincerely appreciate the reviewer's thoughtful feedback and constructive suggestions. We have carefully addressed each point as detailed below, with all changes tracked in the revised manuscript (highlighted in yellow for ease of review).

Dear Authors, Thank you again for your diligent revisions. Your softened causal language, expanded biological discussion, and clearer hypothesis framing have greatly improved the manuscript. I recommend acceptance pending a few minor edits and clarifications, especially around distinguishing mediation from confounding and correcting a mis-applied citation. Please address the points below—organized by line number:

Question 1: Line 51–52 Issue: Wording implies anemia is a subtype of “nutritional deficiencies,” and Ref. 15 does not address anemia.

Fix: Either remove “anemia” from this clause or replace it with a citation specifically linking anemia to sleep disturbances.

Answer: We sincerely appreciate your careful reading. To avoid overgeneralizing the relationship between nutritional deficiencies and sleep disturbances, we have removed “anemia” from the clause in lines 51, 52. Thank you for helping us improve the precision of our manuscript.

Question 2: Lines 106–108 Issue: WHO hemoglobin cutoffs (<12.0 g/dL women; <13.0 g/dL men) omit age-related declines.

Fix: Add a brief note in Methods acknowledging this limitation (and, if feasible, mention a sensitivity analysis with age-specific ranges).

Answer: As suggested, we have added the following acknowledgment in the Methods section (Lines 114, 115):"While WHO cutoffs provide standardized criteria, we acknowledge these do not account for age-related hemoglobin declines."

Question 3: Lines 203–205 Issue: Typo “sleep disturbanced breathing.”

Fix: Correct to “sleep-disordered breathing. ”

Answer: Thank you for pointing out this typographical error. We sincerely apologize for the oversight. As suggested, we have corrected “sleep disturbanced breathing” to “sleep-disordered breathing” in Line 212. This revision ensures accuracy and consistency with standard medical terminology.

Question 4: Lines 206–207 Issue: “Lower anemia” is imprecise.

Fix: Change to “lower hemoglobin levels” or “milder anemia. ”

Answer: Thank you for your insightful comment. We agree that the term "Lower anemia" is imprecise. As suggested, we have revised the text to "lower hemoglobin levels"(Line214).

Question 5: Lines 195–199 & 241–248 Issue: Some definitive causal verbs remain (“mediates,” “could alleviate sleep disturbances”).

Fix: Replace with circumspect phrasing (“may mediate,” “is associated with,” “could reflect”).

Answer: Thank you for your valuable feedback. As suggested, we have revised the language to reflect a more cautious interpretation(Lines239, 256). These changes align with the observational nature of our study and ensure consistency with the available evidence. We appreciate your attention to this important nuance.

Question 6: Numbering and Cross-References Issue: Section/table labels (e.g., S4 vs. S5) and in-text citations sometimes mismatch.

Fix: Ensure every cited figure/table corresponds to the correct label.

Answer: Thank you for your careful review. We have thoroughly checked all section/table labels (e.g., S4, S5) and in-text citations as suggested, and confirmed Confirm that all references in the current version correspond correctly to the tags.If we missed any inconsistencies, please kindly point out the specific instances, and we will promptly correct them. We greatly appreciate your attention to detail.

Question 7: Clarify Mediation vs. Confounding Several passages conflate mediator pathways with confounding influences. Please revise as follows: Iron-Deficiency Mechanism (Lines 206–217): Label body iron status as a confounder, not a mediator, or move to Limitations, noting ferritin wasn’t measured. Inflammatory Pathways (Lines 248–252): Recast inflammation as a shared cause of both reduced grip strength and poor sleep, or explicitly justify how it transmits effects through grip strength. Bidirectional Interplay (Lines 246–249): Choose one primary direction (grip → sleep) for mediation, and present the reverse (sleep → grip) as an alternative hypothesis or effect modifier—not part of the same mediation pathway.

Answer: We sincerely thank the reviewers for their insightful feedback on our analysis. We have made the following revisions to the manuscript to address these issues:

As a response to "Clarify Mediation vs. Confounding". In response, we have revised the text to explicitly frame body iron status as a potential confounder rather than a mediator(Lines 215-217).

As a response to "Inflammatory pathway". In response, we agree that the temporal relationship between inflammation, grip strength, and sleep requires further elaboration. Just as suggested, we have modified the text(Lines 264-268).

As a response to "Bidirectional Interplay". In response, We sincerely appreciate the reviewer's insightful comment regarding the bidirectional interplay between grip strength and sleep quality. Upon careful consideration, we agree that presenting both directions within the same mediation framework could create conceptual ambiguity. In response: We have removed the bidirectional Interplay (original Lines 260-265) to maintain methodological clarity. This modification strengthens the paper's focus while preserving scientific rigor. We thank the reviewer for prompting this important clarification.

Question 8: Muscle Fatigue “Recovery” Mechanism (Lines 249–252)

Current text: “Grip strength reflects overall muscle function, and the anemia-induced weakness can cause increased daytime fatigue, which may necessitate more intense recovery during sleep and potentially disrupt sleep patterns[47].”

Issues: The cited Debevec et al. study examines normobaric hypoxia and bed-rest muscle wasting—environmental oxygen deprivation under inactivity—not anemia-induced hypoxia or sleep architecture. Anemia-related hypoxia co-occurs with other factors (e.g., micronutrient deficiencies, inflammation), making it not equivalent to pure environmental hypoxia.

Fix: Either Provide a sleep-specific citation that directly links anemia-induced muscle fatigue or elevated metabolic demand to disrupted sleep stages, or Flag as speculative, for example: “This remains hypothetical—while environmental hypoxia can exacerbate inactivity-related muscle wasting[Debevec et al. 2018], anemia-related hypoxia (often accompanied by micronutrient deficiencies and inflammation) may not have the same effects on sleep architecture, and this pathway requires direct study.”

Once these edits and clarifications are made, the manuscript will present a coherent, well-justified narrative—appropriately cautious about causality and clear in its distinction between mediators and confounders.

Answer: We sincerely appreciate the reviewer's insightful critique regarding the speculative link between anemia-induced muscle fatigue and sleep disruption. We fully agree that the cited study on normobaric hypoxia (Debevec et al.) does not directly address anemia-related pathways, and we have revised the text to better reflect this distinction. Following the reviewer's second suggested approach, we have: Removed the original speculative statement linking anemia to sleep disruption via muscle recovery. Replaced it with a more cautious formulation that explicitly acknowledges the knowledge gap(Lines 258-263). We believe this revision better aligns with the available evidence while preserving the conceptual framework for future research. Thank you for highlighting this important nuance.

Response to Reviewer #4

Question 1 : I do, however, have a few remaining suggestions that were not fully addressed in your previous response. While my training in Epidemiology typically advises against using cross-sectional data for mediation analysis, I understand that such approaches are commonly employed in the literature to explore associations. Given this context, I would recommend that you explicitly and critically acknowledge this methodological limitation in your study.

Answer: We sincerely appreciate the reviewer's thoughtful comments regarding the methodological considerations of mediation analysis with cross-sectional data. We fully acknowledge the limitations highlighted by the reviewer. To address this limitation, we have: Added a dedicated subsection in the Discussion. The limitations of the research methods were clearly and critically acknowledged. And point out the research direction we aim for in the longitudinal study (Lines 285-290).

Question 2 : Abstract

Methods

Since you do not entail what specific inclusion and exclusion criteria were applied in the abstract, you may just say that 6,057 is your analytic sample. It is by default understood that the analytic sample is a result of certain operation and treatments to the original dataset.

Answer: We appreciate the reviewer’s constructive suggestions. Below are our responses:

As a response to "Analytic sample". In response, we have revised the abstract to state “Following the application of specific inclusion and exclusion criteria, a total of 6,057 Chinese adults aged 60 and above were finally selected as the analysis samples. ” without detailing inclusion/exclusion criteria to save space(Lines 17,18).

As a response to "Variable categories". In response, we have simplified the description of variables in brackets (e.g., sleep [yes/no], anemia [present/absent]) as recommended(Lines 18-20).

As a response to "Regression terminology". In response, we agree that both terms are technically correct. For clarity, we will use “multivariable logistic regression” in the manuscript to emphasize the adjustment for multiple covariates. We thank the reviewer for these thoughtful comments, which have improved the conciseness of our abstract.

Question 3 : Introduction

Please check the PLOS One formatting guidelines if there should be space before [] sign used for in-text citation. If yes, make changes throughout the manuscript.

Answer: We appreciate the reviewer's careful attention to citation formatting. According to PLOS ONE guidelines, we have now ensured that all in-text citations (e.g., [1]) are preceded by a space throughout the manuscript. All instances have been corrected in the revised version.

Question 4: <1st paragraph> While I appreciate the operationalization of "sleep disturbance" in the Methods section, I would respectfully suggest that it might be beneficial to introduce and contextualize the term earlier in the manuscript based on existing literature, ideally in the Introduction. Providing a brief overview of how "sleep disturbance" is defined in the scientific literature—or acknowledging the lack of a standardized definition—could help readers better understand the scope and relevance of your study. Given that "sleep disturbance" can encompass a range of issues such as insomnia, obstructive sleep apnea, and excessive daytime sleepiness, clarifying what is meant in your study from the outset could enhance the clarity and impact of your work. The operational definition fits well under the methods section.

Answer: As you recommended, we have now added a concise definition of "sleep disturbance" in the Introduction section(Lines 38,39). We believe these additions enhance the manuscript’s readability and theoretical grounding. Thank you again for your constructive comment. Please let us know if further clarifications would be helpful.

Question 5 : <4th paragraph> Apologies if I may have overlooked it, but I did not clearly see the theoretical foundation referenced in lines 75–78. If it is present, it may benefit from clearer articulation. Additionally, as a general convention, the theoretical framework is typically introduced before the last paragraph in the introduction section where you describe the research objective. Also, ideally the conceptual framework positioned after the research objectives and before the Methods section. Presenting it there can help anchor the study's rationale and guide readers through the analytical approach more effectively. I would recommend using conceptual framework than DAGS since your study is cross-sectional.

Answer: We sincerely appreciate the reviewer’s thoughtful suggestions to improve the manuscript’s clarity and structure. As suggested, we added the "conceptual framework" after the research objective and before the Methods(Lines 78-80,S1 Fig). We hope the revised manuscript addresses the reviewer’s concerns effectively.

Question 6: Methods <1st paragraph> If CHARLES is a longitudinal study, not a part of longitudinal study, then please correct the statement where it means that CHARLES is a part of a longitudinal study.

Answer: Thank you for your careful reading and valuable feedback.We sincerely appreciate your attention to detail, which has helped us improve the accuracy of our manuscript. We apologize for any confusion caused by the original phrasing. To clarify: CHARLS is a longitudinal study. We have revised the Methods section (Lines 83, 84) to explicitly state: "CHARLS is a longitudinal survey that represents the national population of Chinese individuals aged 45 and above living in communities.". We thank the reviewer for highlighting this important clarification. The revised text now accurately reflects the study design. Please let us know if further adjustments are needed.

Question 7: Given that the CHARLS dataset is longitudinal, your decision to rely solely on Wave 3 for a cross-sectional analysis warrants a clear and well-reasoned justification—particularly in light of your acknowledgment that research on the causal relationship between anemia and sleep disturbance remains limited. What distinguishes your cross-sectional study from existing ones that have already explored this association? Does your work offer new insights, or does it largely replicate previously established findings? Additionally, it is important to address the limitations inherent in using cross-sectional data to test mediating relationships. Such analyses are generally viewed as exploratory rather than confirmatory, due to the inability to establish temporal ordering and causal direction. Given you already have access to longitudinal data, why motivates you to prefer a cross-sectional analysis? Please, clarify the rationale behind focusing exclusively on a single wave for establ

---

## [Decision Letter · Decision Letter 2]

5 Aug 2025

Dear Dr. Zhao,

Thank you for submitting your manuscript to PLOS ONE. After careful consideration, we feel that it has merit but does not fully meet PLOS ONE’s publication criteria as it currently stands. Therefore, we invite you to submit a revised version of the manuscript that addresses the points raised during the review process.

Please make peer-to-peer modifications to the reviewer's comments.

We look forward to receiving your revised manuscript.

Kind regards,

Qian Wu

Academic Editor

PLOS ONE

Journal Requirements:

Reviewers' comments:

Reviewer's Responses to Questions

**Comments to the Author**

Reviewer #1: All comments have been addressed

Reviewer #4: All comments have been addressed

2. Is the manuscript technically sound, and do the data support the conclusions?

Reviewer #1: Partly

Reviewer #4: Yes

3. Has the statistical analysis been performed appropriately and rigorously?

Reviewer #1: Yes

Reviewer #4: Yes

4. Have the authors made all data underlying the findings in their manuscript fully available?

Reviewer #1: Yes

Reviewer #4: Yes

5. Is the manuscript presented in an intelligible fashion and written in standard English?

Reviewer #1: Yes

Reviewer #4: Yes

Reviewer #1: My Reviewer Comment (R1)

To the Editor and Authors,

Thank you for the thorough revision of the manuscript and for the detailed responses to my previous comments. I would like to begin by acknowledging that the paper has been significantly improved through the revision process. The more coherent introduction, the tempered causal language, and the engagement with specific critiques (such as removing the bidirectional hypothesis) are commendable and have substantially enhanced the quality of the work.

The manuscript now presents a compelling statistical analysis, and I recommend it for acceptance pending one final but critical revision to the Discussion section.

The core issue that persists after this revision is the conceptual conflation of mediation and confounding when discussing the physiological mechanisms (lines 251-268). While the authors have attempted to address this, the proposed mechanisms still do not consistently describe a plausible mediational pathway. Instead, they introduce alternative causal models that inadvertently undermine the paper's central thesis of mediation.

Specific Points for Revising the Discussion:

To illustrate the issue precisely, I will analyze the three mechanisms currently proposed in the text (lines 252-268):

1. Mechanism: "Muscle fatigue and recovery" (Lines 252-258)

Assessment: Well-revised. This is the only mechanism that is logically structured as a mediational pathway (Anemia → Muscle Function/Fatigue → Sleep Disturbance). The authors have commendably addressed my previous critique (Q8) by flagging the link as "speculative" and acknowledging the limitations of the cited hypoxia model. This section now stands as a good example of scientific caution.

2. Mechanism: "Inflammatory pathways" (Lines 258-262)

Assessment: The core problem remains unresolved. This section remains logically flawed and contradicts the mediation hypothesis.

Lines 259-260: The statement that inflammation may be a "shared underlying mechanism" for both reduced grip strength and poor sleep is the textbook definition of confounding, not mediation. A confounder (inflammation) is a common cause that can create a statistical association between two variables without a direct causal link between them.

Lines 261-262: The alternative, that inflammation impairs muscle function which in turn disrupts sleep (Inflammation → Grip Strength → Sleep), describes a different causal pathway that does not explain the role of anemia as the initial exposure.

Conclusion: This section actively argues against the paper's central claim. Despite changes in wording, the fundamental logical confusion has not been corrected.

3. Mechanism: "Energy metabolism" (Lines 262-264)

Assessment: Logically insufficient as a mechanism for mediation. This argument describes the direct effect of anemia on the outcome (A→B) by stating that reduced oxygen delivery impairs "daytime physical function and recovery." It fails to articulate the specific, necessary intermediary role of handgrip strength (M) in this causal chain. As written, the point is too vague and restates the study's initial premise rather than explaining the mechanism of mediation.

Recommendation for Final Revision:

To bring this manuscript to a publishable standard, I propose a clear structural solution that resolves these logical issues and enhances its scientific rigor:

Retain Mechanism 1 ("Muscle fatigue") as the sole hypothesized mediational pathway, continuing to emphasize its speculative nature.

Remove the "Inflammatory pathways" and "Energy metabolism" sections from the list of mediational mechanisms.

Instead, create a new, separate paragraph in the Discussion, for instance, under the heading "Alternative Interpretations and Potential Confounding."

In this new paragraph, place the argument regarding inflammatory pathways, explicitly framing it as a major potential confounder that could also explain the observed associations.

Also in this paragraph, discuss how "impaired energy metabolism" could represent a direct physiological effect of anemia on sleep, operating in parallel with the statistical mediation observed.

Why this change is crucial:

By restructuring the discussion in this manner, the authors will transform a logical weakness into a scientific strength. Instead of incorrectly labeling confounding as mediation, they will demonstrate a sophisticated understanding of causal inference. This shows they recognize the complexity of the topic and the inherent limitations of their cross-sectional design. Such a change would place their interpretation on a much more solid foundation, making the paper's conclusions far more robust and credible.

Reviewer #4: Thank you for your thoughtful revisions. I appreciate the time and effort you have taken to address my comments. I hope my feedback has been helpful in improving the quality of your work.

**Do you want your identity to be public for this peer review?** For information about this choice, including consent withdrawal, please see our Privacy Policy

Reviewer #1: **Yes: ** Dr. Thomas C. Carmine

Reviewer #4: **Yes: ** Aman Shrestha

---

## [Decision Letter · Decision Letter 3]

18 Aug 2025

Dear Dr. Zhao,

Thank you for submitting your manuscript to PLOS ONE. After careful consideration, we feel that it has merit but does not fully meet PLOS ONE’s publication criteria as it currently stands. Therefore, we invite you to submit a revised version of the manuscript that addresses the points raised during the review process.

Please make peer-to-peer modifications to the reviewer's comments.

We look forward to receiving your revised manuscript.

Kind regards,

Qian Wu

Academic Editor

PLOS ONE

Journal Requirements:

Reviewers' comments:

Reviewer's Responses to Questions

**Comments to the Author**

Reviewer #1: (No Response)

2. Is the manuscript technically sound, and do the data support the conclusions?

Reviewer #1: Partly

3. Has the statistical analysis been performed appropriately and rigorously?

Reviewer #1: Yes

4. Have the authors made all data underlying the findings in their manuscript fully available?

Reviewer #1: Yes

5. Is the manuscript presented in an intelligible fashion and written in standard English?

Reviewer #1: Yes

Reviewer #1: To the Editor and Authors,

Thank you for the revised manuscript and for the thoughtful response letter.

I am recommending Minor Revision because I believe the final changes required are straightforward to implement. However, this recommendation is strictly conditional on resolving the core conceptual issue that, despite the recent revisions, has persisted from my previous review.

In your response letter, you articulated your rationale for the current approach, citing a desire to preserve "theoretical integrity" and "avoid the potential omission of biological mechanisms." I fully agree that these mechanisms are important. The crucial point that was missed in this revision, however, is that these mechanisms must be framed with correct causal terminology to ensure the paper's logical integrity.

The central issue remains: the manuscript continues to conflate mediation with confounding. For example, describing how inflammation affects both grip strength and sleep independently is a description of confounding, not mediation. Presenting it as a mediating pathway is a significant logical flaw that undermines the paper's main conclusion.

To resolve this persistent issue once and for all, I must insist on a specific structural change. This is not a suggestion, but a requirement for my recommendation of acceptance.

Required Action:

Clearly designate "Muscle fatigue and recovery" as the primary hypothetical mediating pathway.

Create a new, separate paragraph with the heading: "Alternative Explanations and Potential Confounding".

Move your important discussions of "inflammatory pathways" and "energy metabolism" into this new paragraph. You must then reframe them using precise causal language. For example:

For Inflammation: State that it may act as a common cause or confounder.

For Energy Metabolism: Explain that it could represent a confounding pathway or a direct effect of anemia.

Why This Non-Negotiable Revision is Necessary:

It Corrects the Persistent Logical Flaw: This structure finally distinguishes between your proposed mediation hypothesis and its most likely alternatives.

It Aligns with Scientific Rigor: Correctly labeling potential confounders is a sign of a robust scientific argument and is a best practice in causal inference.

It Provides a Clear Path to Acceptance: This offers a simple, actionable solution to the only major issue remaining in an otherwise improved manuscript.

I trust that the authors will now understand the necessity of this change. I look forward to reviewing a final version that incorporates these specific revisions and will be pleased to recommend acceptance at that time.

**Do you want your identity to be public for this peer review?** For information about this choice, including consent withdrawal, please see our Privacy Policy

Reviewer #1: **Yes: ** Dr. Thomas C. Carmine

---

## [Author Response · Author response to Decision Letter 4]

19 Aug 2025

Response to Reviewer #1

Dear Dr. Carmine TC

Thank you very much for your letter and the comments about our paper “The Relationship between anemia and sleep disturbances among Older Adults in China: The Mediating Role of handgrip strength (Manuscript ID: 517084bef51b020d)”. After carefully studying the reviewer' comments and advice, we have made corresponding changes to the paper. Revised sections are identified with red text in the paper.

There are no conflicts of interest regarding this work. All authors have read the revised manuscript and approved its submission to the PLOS ONE. Please do not hesitate to contact us if we can be of any further assistance.

Thank you and best regards.

Yours Sincerely

Xiaojiang Zhao

mail: zhaoxiaojiang2010@163.com

Response to Reviewer #1

Question: Thank you for the revised manuscript and for the thoughtful response letter. I am recommending Minor Revision because I believe the final changes required are straightforward to implement. However, this recommendation is strictly conditional on resolving the core conceptual issue that, despite the recent revisions, has persisted from my previous review.

In your response letter, you articulated your rationale for the current approach, citing a desire to preserve "theoretical integrity" and "avoid the potential omission of biological mechanisms." I fully agree that these mechanisms are important. The crucial point that was missed in this revision, however, is that these mechanisms must be framed with correct causal terminology to ensure the paper's logical integrity.

The central issue remains: the manuscript continues to conflate mediation with confounding. For example, describing how inflammation affects both grip strength and sleep independently is a description of confounding, not mediation. Presenting it as a mediating pathway is a significant logical flaw that undermines the paper's main conclusion.

To resolve this persistent issue once and for all, I must insist on a specific structural change. This is not a suggestion, but a requirement for my recommendation of acceptance.

Required Action:

Clearly designate "Muscle fatigue and recovery" as the primary hypothetical mediating pathway.

Create a new, separate paragraph with the heading: "Alternative Explanations and Potential Confounding".

Move your important discussions of "inflammatory pathways" and "energy metabolism" into this new paragraph. You must then reframe them using precise causal language.

For example:

For Inflammation: State that it may act as a common cause or confounder.

For Energy Metabolism: Explain that it could represent a confounding pathway or a direct effect of anemia.

Why This Non-Negotiable Revision is Necessary:

It Corrects the Persistent Logical Flaw: This structure finally distinguishes between your proposed mediation hypothesis and its most likely alternatives.

It Aligns with Scientific Rigor: Correctly labeling potential confounders is a sign of a robust scientific argument and is a best practice in causal inference.

It Provides a Clear Path to Acceptance: This offers a simple, actionable solution to the only major issue remaining in an otherwise improved manuscript.

I trust that the authors will now understand the necessity of this change. I look forward to reviewing a final version that incorporates these specific revisions and will be pleased to recommend acceptance at that time.

Answer: We sincerely thank you for your comments on the "intermediary role and confounding factors" issue in our manuscript as well as for your constructive suggestions. Your insights have been invaluable in strengthening the logical rigor and clarity of our work. We have carefully implemented all the requested changes as outlined below:

1). As you rightly pointed out, we have: Clearly designated "Muscle fatigue and recovery" as the primary mediating pathway in the revised manuscript (Lines 258,259).

2).Created a new subsection titled "Alternative Explanations and Potential Confounding" (Lines 266,278). Where we now discuss: Inflammatory pathways as potential common causes/confounders. Energy metabolism as a potential confounding pathway.

Once again, we would like to express our gratitude for the valuable time and professional guidance you have provided in enhancing the quality of this article. Please let us know if further adjustments are needed.

---

## [Decision Letter · Decision Letter 4]

18 Sep 2025

The relationship between anemia and sleep disturbances among older Chinese adults: The mediating role of handgrip strength

PONE-D-25-13112R4

Dear Dr. Zhao,

We’re pleased to inform you that your manuscript has been judged scientifically suitable for publication and will be formally accepted for publication once it meets all outstanding technical requirements.

Kind regards,

Qian Wu

Academic Editor

PLOS ONE

Additional Editor Comments (optional):

Reviewer #1:

Reviewers' comments:

Reviewer's Responses to Questions

**Comments to the Author**

Reviewer #1: All comments have been addressed

2. Is the manuscript technically sound, and do the data support the conclusions?

Reviewer #1: Yes

3. Has the statistical analysis been performed appropriately and rigorously?

Reviewer #1: Yes

4. Have the authors made all data underlying the findings in their manuscript fully available?

Reviewer #1: Yes

5. Is the manuscript presented in an intelligible fashion and written in standard English?

Reviewer #1: Yes

Reviewer #1: To the Editor and Authors,

I am pleased to recommend this manuscript for acceptance.

The authors have successfully resolved all previous concerns. The creation of the new section, "Alternative Explanations and Potential Confounding," and the correct reframing of the inflammatory and metabolic pathways have fixed the core conceptual issue.

The manuscript is now scientifically rigorous and logically sound. It is a valuable contribution, and I have no further reservations about its publication.

**Do you want your identity to be public for this peer review?** For information about this choice, including consent withdrawal, please see our Privacy Policy

Reviewer #1: **Yes: ** Dr. Thomas C. Carmine

---

## [Editor Report · Acceptance letter]

PONE-D-25-13112R4

PLOS ONE

Dear Dr. Zhao,

I'm pleased to inform you that your manuscript has been deemed suitable for publication in PLOS ONE. Congratulations! Your manuscript is now being handed over to our production team.

Kind regards,

on behalf of

Dr. Qian Wu

Academic Editor

PLOS ONE